# VGRCOT: a one-tube visual detection method for group B *Streptococcus* combining RPA and CRISPR/Cas12a for point-of-care testing in reproductive health

Caixia Ji,[1] Liqiang Ru,[2,3] Tiao Han,[4] Gang Mai,[5] Laibao Zheng,[4] Yayun Jiang[5]

**ABSTRACT**  Group B *Streptococcus* (GBS) is a significant pathogen that causes perinatal infections, seriously threatening the health of pregnant women and newborns. Prophylactic antibiotic treatment for pregnant women who screen positive for GBS can notably reduce the incidence and fatality of neonatal infections. Herein, we developed a visual nucleic acid method for GBS that integrates RPA and CRISPR/Cas12a in a one-tube setup, termed VGRCOT. The VGRCOT method achieved one-tube detection by adding the appropriate reagents to the bottom and lid of the EP tube, respectively. By rigorous optimization of ssDNA-FQ reporter concentration, crRNA concentration, RPA reaction time, and CRISPR/Cas12a cleavage time, VGRCOT can exhibit fluorescence under ultraviolet light, enabling visual detection. Under optimal conditions, VGRCOT has a satisfactory selectivity, and the detection limit was determined as $10^1$ copies/reaction. Finally, VGRCOT also showed good performance comparable to qPCR in the actual detection of clinical specimens. Due to its ease of operation and convenient signal acquisition, VGRCOT shows promise for point-of-care testing in reproductive health.

**IMPORTANCE**  This study presents a convenient, sensitive, and accurate visual detection method (VGRCOT) for GBS, combining RPA and CRISPR/Cas12a in a single reaction vessel. Through optimization of experimental conditions, VGRCOT enables detection within 60 min, with a minimum detection limit of $10^1$ copies per reaction. VGRCOT offers several advantages by adding the appropriate reagents to the bottom and lid of the EP tube. The one-tube visualization method effectively prevents aerosol contamination, simplifies procedures, and enables visual detection without complex instruments, making it ideal for resource-limited environments. Additionally, its editable crRNA and the use of commonly available laboratory reagents allow for easy reprogramming to detect various pathogens, supporting scalable and low-cost batch production.

**KEYWORDS**  CRISPR/Cas12a, visual detection, one-tube, GBS, POC testing

Group B *Streptococcus* (GBS), also known as *Streptococcus agalactiae*, is a gram-positive bacterium that inhabits the human reproductive and gastrointestinal tracts (1). GBS is a significant pathogen for newborns and pregnant women, presenting a serious threat to maternal and neonatal health. An effective measure to prevent perinatal infections is to screen all pregnant women between 35 and 37 weeks of gestation for GBS carriage to proactively provide antibiotics to carriers (2). Hence, a convenient, visual, and without need complicated instruments method for GBS is crucial to protect the life and health of the mother and newborn.

Currently, bacterial culture, immunoassay, and polymerase chain reaction (PCR) are commonly used for the clinical detection of GBS (3, 4). Although bacterial culture is considered the gold standard for diagnosing GBS, it is cumbersome and

**Peer Reviewers** Reza Azizian, Ilam University of Medical Sciences, Ilam, Iran; Tahereh navidifar, Ahvaz Jundishapur University of Medical Sciences, Ahvaz, Iran

Address correspondence to Gang Mai, maigang68@hotmail.com, Laibao Zheng, zhenglaibao@wmu.edu.cn, or Yayun Jiang, dyjiangyayun@163.com.

Caixia Ji and Liqiang Ru contributed equally to this article. Author order was determined by drawing straws.

The authors declare no conflict of interest.

See the funding table on p. 11.

time-consuming. Immunoassays, which rely on specific antibody–antigen interactions, are rapid but have low sensitivity. PCR offers high sensitivity and specificity; however, it requires sophisticated equipment, skilled personnel, and a laboratory environment. Several FDA-approved molecular diagnostic tests for GBS have been developed in recent years (5). One example is the Alethia GBS DNA Amplification Assay, which utilizes loop-mediated isothermal amplification (LAMP) technology. Another is the Solana GBS assay, an *in vitro* diagnostic test that employs helicase-dependent amplification (HDA) for the direct, qualitative detection of GBS from vaginal or rectal swabs after 18–24 h of incubation in carrot broth cultures. However, these methods also have certain limitations in point-of-care (POC) testing.

POC testing is critical in monitoring infectious diseases, tracking disease progression, and responding to outbreaks. It is especially valuable in resource-limited settings and offers significant advantages for public health strategies (6). Various POC testing methods have been developed, with the most notable including lateral flow assays (LFA), biosensor-based assays, and clustered regularly interspaced short palindromic repeats (CRISPR)-based assays (7). LFA provides a rapid, cost-effective, and portable approach suitable for initial screening. However, its relatively low sensitivity and specificity reduce diagnostic accuracy, especially when sampling quality is suboptimal (8). Biosensor-based assays offer high sensitivity and specificity with a portable design, but biorecognition components can be costly, and environmental factors may influence the results (9).

CRISPR/CRISPR-associated protein (Cas) systems are acquired immune systems formed by bacteria and archaea during biological evolution to defend against invasion by exogenous substances such as phages and plasmids. The CRISPR/Cas system has the function of accurately identifying and cleaving target DNA or RNA sequences, serving as a crucial tool for transcriptional regulation, gene editing, and DNA/RNA detection, acclaimed as the next-generation molecular detection technology (10, 11). Notably, CRISPR/Cas12a can not only specifically identify and cleave target double-stranded DNA (dsDNA) but also indiscriminately cleave single-stranded DNA (ssDNA), demonstrating significant application potential in the POC testing (12). However, several limitations may affect its practicality, including low sensitivity, technical complexity, and risks of false positives due to contamination, all of which hinder its practical application in rapid and low-resource settings (7). Recombinase polymerase amplification (RPA) is an amplification technique that mimics nucleic acid replication processes *in vitro*, utilizing recombinase, single-stranded DNA-binding proteins, and strand-displacement DNA polymerase (13). RPA offers high sensitivity, excellent specificity, and easy operation as key advantages, which are widely used in biosensing and are an ideal choice for developing CRISPR/Cas-based methods (14). Some fluorescent assays based on CRISPR/Cas12a and RPA have been constructed, but these mostly require sophisticated instrumentation to collect fluorescent signals and are realized by two separate reactions. Furthermore, the wo-tube method involves uncapping and liquid transfer, exposing the RPA amplification product to air, which increases the risk of laboratory contamination and potentially leads to false positives (15).

In this study, a visual detection method for GBS was developed using RPA and CRISPR/Cas12a within a single reaction vessel, adding the appropriate reagents to the bottom and lid of the EP tube. Numerous target genes were amplified by carefully designing and selecting RPA primers. By meticulously optimizing the ssDNA-FQ reporter concentration, crRNA concentration, RPA reaction time, as well as CRISPR/Cas12a cleavage time, fluorescence can be observed under UV light for visual detection. The detection process is conducted entirely in a single tube, necessitating only short periods of centrifugation and incubation. The VGRCOT method is convenient, sensitive, and accurate, with a minimum detection limit of $10^1$ copies/reaction. This portable method reduces false positives due to aerosol contamination and expands the application scenarios of GBS detection.

## MATERIALS AND METHODS

### Reagents

LbCas12a (Catalog # M0653T) and RNA transcription Kits (Catalog # E2050S) were purchased from New England Biolabs Inc. (United States). RAA FAST (Catalog # B00000), the recombinase polymerase amplification (RPA) nucleic acid amplification kits were obtained from Qitian Gene Biological Co., Ltd. (Jiangsu, China). RNA purification kits (Catalog # R0071S) were purchased from Biyuntian Biotechnology Co. (Shanghai, China). Ultra GelRed (Catalog # GR501-01) and a 100 bp DNA Ladder (Catalog # MD104-01) were provided by Vazyme Co., Ltd. (Nanjing, China). SGExcel Fast SYBR qPCR Mixture (Catalog # B630011), crRNA, and other nucleic acid sequences were synthesized by Sangon Biotech (Shanghai, China). *Escherichia coli* (*E. coli*, ATCC 25922), *Pseudomonas aeruginosa* (*P. aeruginosa*, ATCC 27853), *Staphylococcus epidermidis* (*S. epidermidis*, ATCC 12228), *Staphylococcus aureus* (*S. aureus*, ATCC 29213), and *Streptococcus agalactiae* (*S. agalactiae*, ATCC 12386) were laboratory-preserved strains.

### Preparation of target DNAs

GBS genomic DNA was extracted using the Bacterial Genome Extraction Kit (TaKaRa), and nucleic acid targets were amplified from the genomic DNA template using Ex Taq DNA Polymerase (TaKaRa). The products were purified by FastPure Gel DNA Extraction Mini Kit, and the concentration was determined by ultra-micro spectrophotometer (DeNovix) and stored at −20℃ for backup.

### Preparation of crRNAs

In total, this section includes crRNA design, template synthesis, and crRNA transcription and purification. The *cfb* gene sequence was utilized as the target sequence for designing the crRNA sequence, ensuring no overlap between the crRNA recognition site and the primer sequence, and selecting a site containing the PAM (TTTN) motif. The crRNA template strand and its complementary strand were synthesized and purified by Shanghai Sangon Bioengineering Company Limited. The crRNA template strand was mixed at equal concentrations with its complementary strand, and the template dsDNA was prepared by the annealing method. The crRNA was synthesized in strict accordance with the instructions of HiScribe T7 Quick High Yield RNA Synthesis Kit (NEB). In brief, the transcription system was first incubated at 37℃ for 16 h. Then, DNase I was added, and the DNA was digested by incubation at 37℃ for 60 min, and DNase I was inactivated by incubation at 65℃ for 15 min. The process of crRNA purification was carried out strictly according to the BeyoMag RNA Clean Magnetic Beads Purification Kit using magnetic beads. The transcribed crRNA was purified and assessed using a 4% agarose gel to judge the transcripts.

### CRISPR/Cas12a trans-cleavage assay

The CRISPR/Cas12a *trans* cleavage reaction system contained 2 µL Cas12a (1 µM), 2 µL crRNA (1 µM), 4 µL ssDNA-Probe (1 µM), 2.5 µL 10 × NE Buffer, and 12.5 µL ddH$_2$O. The components were mixed thoroughly and incubated for 40 min at 37℃ in a constant metal bath. The visualized fluorescence results were observed under UV light and then transferred to a 384-well plate to measure the fluorescence spectrum by a BioTek SynergyNeo2 Microplate Reader (Agilent, USA).

### Feasibility validation of a one-tube assay for the detection of GBS

The VGRCOT one-tube assay consists of two parts: the RPA system and the CRISPR/Cas12a cleavage system. Ten microliters of RPA premix and 5 µL of $10^6$ CFU/mL of GBS cleavage template were mixed and added to the bottom of the EP tube. The CRISPR system consisted of 2 µL of Cas12a (1 µM), 2 µL of crRNA (1 µM), 4 µL of ssDNA-Probe (1 µM), and 2.5 µL of 10 × NE Buffer and was mixed well. The CRISPR premix was then

added to the lid of the EP tube. The CRISPR/Cas12a reaction system includes a concentrated NE Buffer, a viscous solution with a relatively high surface tension that adheres well to tube caps. After 30 min of reaction at 37°C in a constant metal bath, the CRISPR at the top of the EP tube was centrifuged to the bottom with a palm centrifuge, mixed well, and the reaction was continued for 40 min at 37°C in a constant metal bath. At the end of the reaction, the tube was placed in a UV light to observe the fluorescence intensity and was transferred to a 384-well plate, and the fluorescence spectra and its concentration at 522 nm were measured by a multifunctional enzyme labeling instrument, spectrum, and its fluorescence value at 522 nm.

## Bacterial culture

GBS was inoculated in a blood agar plate and incubated at 37°C overnight. A single colony was then picked and inoculated in a BHI liquid medium for bacterial enrichment (37°C, 250 rpm/min). The bacteria were enriched by centrifugation and washed twice with ultrapure water, and then the bacterial concentration was approximated by adjusting the optical density of the bacterial suspension with an absorbance of 1.0 at a wavelength of 600 nm. The bacterial suspension was diluted by gradient dilution, and a more accurate bacterial concentration was obtained by the plate counting method. Genomic DNA from each group of bacteria was extracted via thermal lysis for the VGRCOT DNA detection performance evaluation experiments. The obtained nucleic acid extracts were stored at −20°C for use.

## qPCR assay

The qPCR assay for the *cfb* gene was performed on an ABI QuantStudio 5 (Applied Biosystems, Thermo Fisher Scientific, Waltham, MA, USA) according to the instructions of the SGExcel FastSYBR qPCR Mixture. Specifically, the qPCR mixtures contained 10 µL of 2 × SGExcel FastSYBR Mixture, 0.4 µL of each primer (10 µM), 5 µL of DNA template, and 4.2 µL of RNase-Free ddH$_2$O. The reaction condition was 95°C for 30 s, and 40 cycles of 95°C for 5 s and 60°C for 30 s.

## RESULTS AND DISCUSSION

### Principle of the VGRCOT method

As shown in Fig. 1, the VGRCOT method consists of two parts: the RPA system located at the EP tube's bottom and the CRISPR/Cas12a system on the inside of the cover. Initially, incubate the EP tube in a 37°C metal bath for 20 min. Then, employ a handheld centrifuge to settle the CRISPR system at the tube's base, thoroughly mix it, and maintain the reaction at a constant 37°C in the metal bath for 30 min. In the absence of the target DNA, the RPA reaction fails to amplify, leaving the ssDNA probe intact and without fluorescence. Conversely, in the presence of the target gene, a substantial quantity of the target gene can be rapidly amplified following the RPA reaction. Upon the integration of the RPA reaction system with the CRISPR/Cas12a system, the CRISPR/Cas12a system was activated. The trans-cleavage activity of CRISPR/Cas12a is activated, leading to the cleavage of the ssDNA-probe and generating green fluorescence visible to the naked eye under ultraviolet light.

The CAMP factor is a hemolysin specifically secreted by GBS. We selected the *cfb* gene (GenBank accession No GU217532) that encodes the CAMP factor as the target gene for nucleic acid diagnosis of GBS-specific genes. CrRNA sequences for the *cfb* gene were used according to previous literature reports listed in Table S1 (16). The digestion of the crRNA transcript by DNase I and its subsequent analysis by gel electrophoresis. As shown in Fig. S1a, distinct single electrophoresis band confirms the successful synthesis of crRNA. The trans-cleavage activity of crRNA-guided Cas12a was verified by ssDNA-Probe (Table S1). As shown in Fig. S1b-c, significant fluorescence is only produced when all components are present in the detection system. The same visual detection results were obtained by naked-eye observation of fluorescent images under UV light (Fig. S1d).

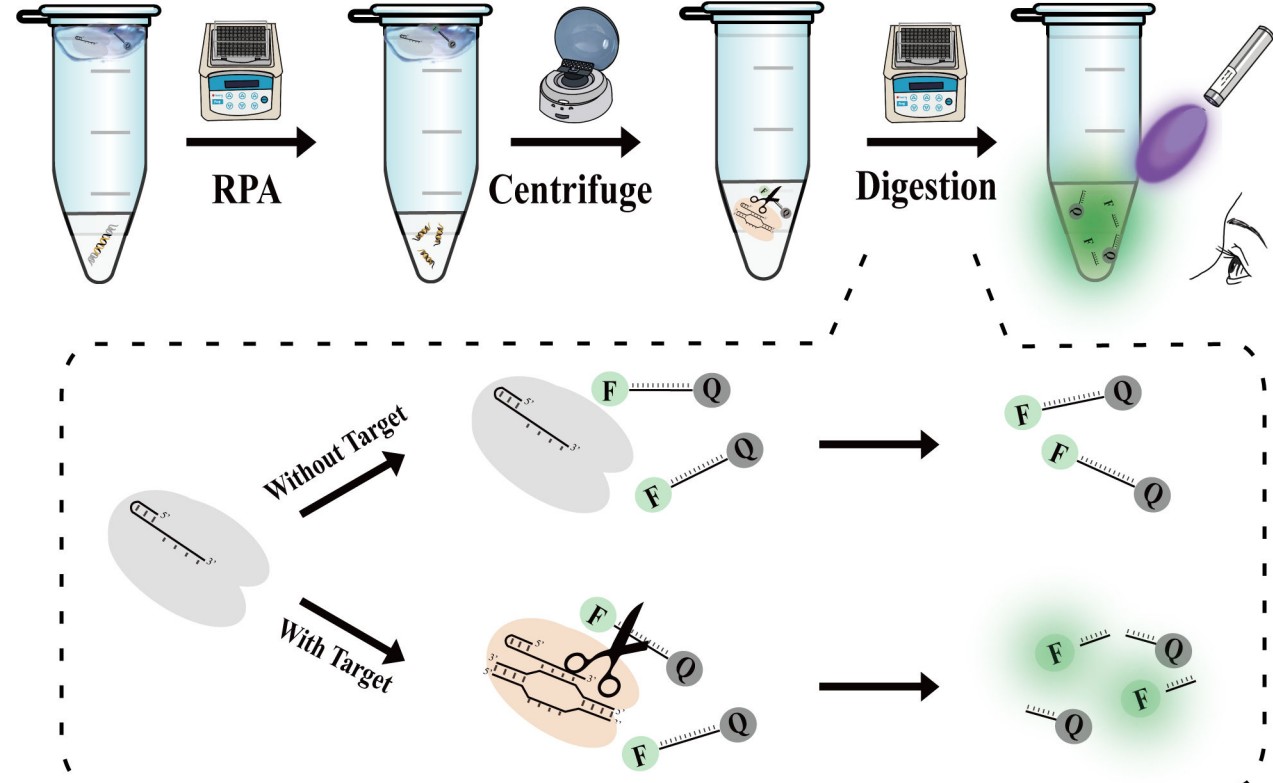

**FIG 1** Schematic illustration of the VGRCOT for visual detection.

## Feasibility analysis of the VGRCOT method

We designed two pairs of RPA primers and analyzed their amplification efficiency by gel electrophoresis. As shown in Fig. S2, the intensity of the amplified band amplified by primer pair 1 is significantly higher than that of pair 2, so it was selected for subsequent experiments. We added the RPA reaction system to the bottom of the EP tube, and the CRISPR/Cas12a system was gently added to the cap of the tube. The mixture was then incubated at 37°C for 20 min. Subsequently, the two reaction systems were merged by centrifugation, and the incubation was continued for 30 min. Upon completion of the reaction, the fluorescence intensity was observed under UV light, realizing the one-tube detection of GBS (Fig. 2a). As shown in Fig. 2b and c, when all the reaction components were present, a distinct peak at 522 nm was visible, and obvious fluorescence was visible by the naked eye under UV light (Fig. 2d). This proves the feasibility of the VGRCOT method.

## Optimization of the reaction parameters

To ensure well detection performance, we optimized the ssDNA-FQ reporter concentration, crRNA concentration, RPA reaction time, and CRISPR/Cas12a cleavage time. Initially, various concentrations of crRNA were tested to assess their effects on the fluorescence signal intensity and visualization detection performance of VGRCOT. As shown in Fig. 3a and b, the fluorescence intensity was enhanced with the increase of crRNA concentration and reached a plateau at 1 µM. This result was also validated through visual inspection under UV light, revealing prominent green fluorescence in the 1 µM group (Fig. 3c). The ssDNA-FQ reporter concentration directly affects the visualization detection effect. Notably, a higher error margin was observed at 2 µM, with visualization results showing fluorescence signals stabilizing above 1 µM. Consequently, the 1 µM group was chosen for further testing (Fig. 3d through f). To reduce detection time without compromising sensitivity, we optimized the CRISPR/Cas12a cleavage time and RPA reaction duration.

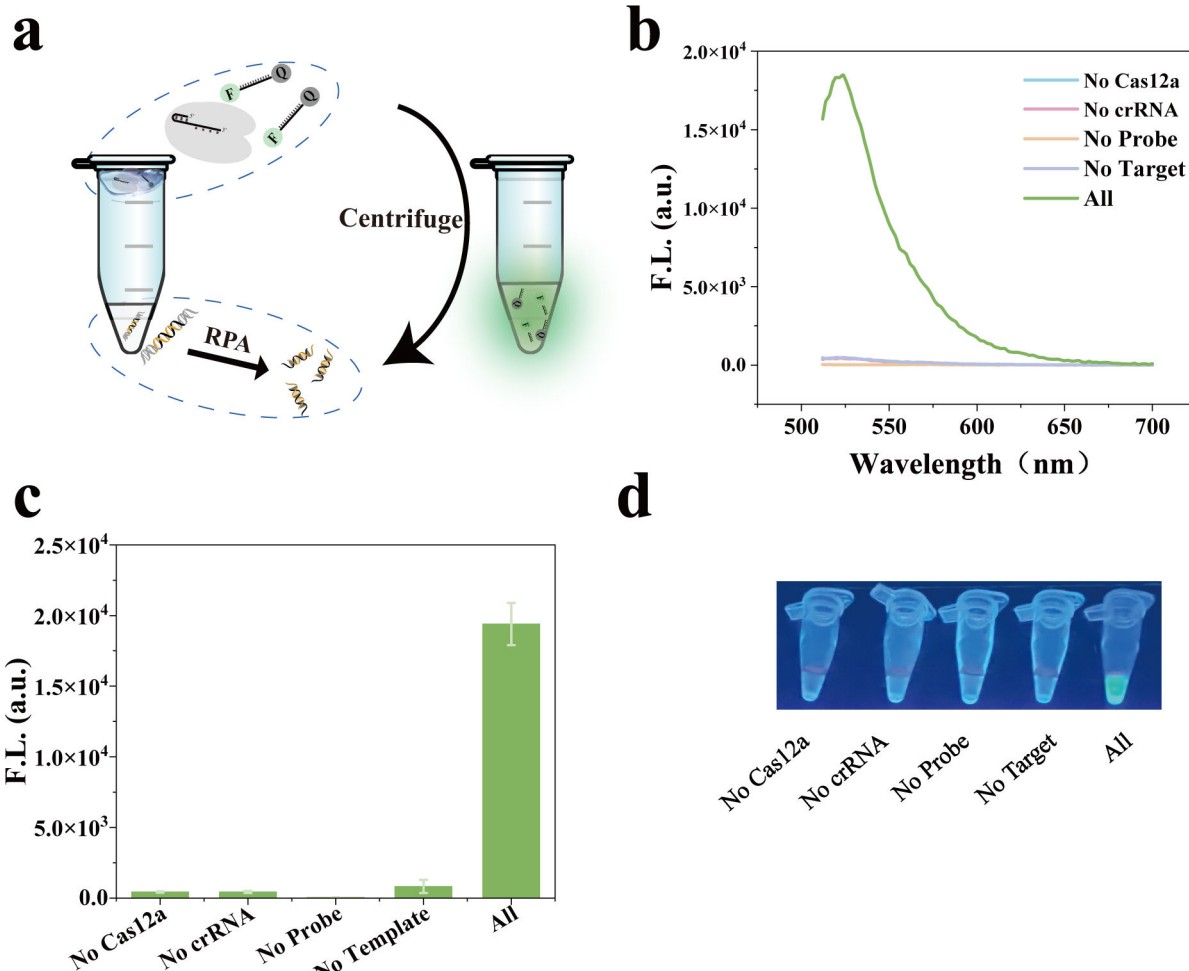

**FIG 2** Feasibility analysis of VGRCOT method. (a) Schematic illustration of the VGRCOT method operation. (b) Fluorescence spectra demonstrate the feasibility of the VGRCOT method. (c) A histogram of fluorescence intensity at 522 nm verifies the feasibility of the VGRCOT method. (d) Representative photographs for the feasibility analysis of the VGRCOT method under UV light.

Fig. 3g shows the fluorescence dynamic change curves of each group with the trans-cleavage activity of CRISPR/Cas12a after RPA reacted for different times, respectively. Fig. 3h and i show the histogram of fluorescence values and photos under UV after 40 min of CRISPR/Cas12a cutting. In Fig. 3g, as the CRISPR/Cas12a reaction time is prolonged, the fluorescence intensity increases accordingly and slows down after 40 min. Consequently, to control the detection time, a 20 min RPA amplification time and a 40 min CRISPR/Cas12a reaction time were taken as the optimal response parameters for the subsequent experiments.

### Performance of the VGRCOT method for GBS detection

Under optimal conditions, we evaluated the specificity and sensitivity of the VGRCOT method in detecting GBS. Initially, we calculated the cut-off value of the VGRCOT method by multiple replicates without using GBS (Fig. S3). The threshold was calculated using the formula TH = $X$ + 3 SD, where $X$ is the mean signal intensity and SD is the corresponding standard deviation (9). Specifically, TH = 783.6 + 3 ×80.73 = 1025.79. Therefore, the threshold value of 1025.79 is used to distinguish between positive and negative results. Four control strains (*E. coli*, *P. aeruginosa*, *S. epidermidis*, and *S. aureus*), along with a mixture of these strains and GBS, were selected to explore the selectivity and anti-interference ability of the VGRCOT method in GBS detection. As shown in Fig. 4a through

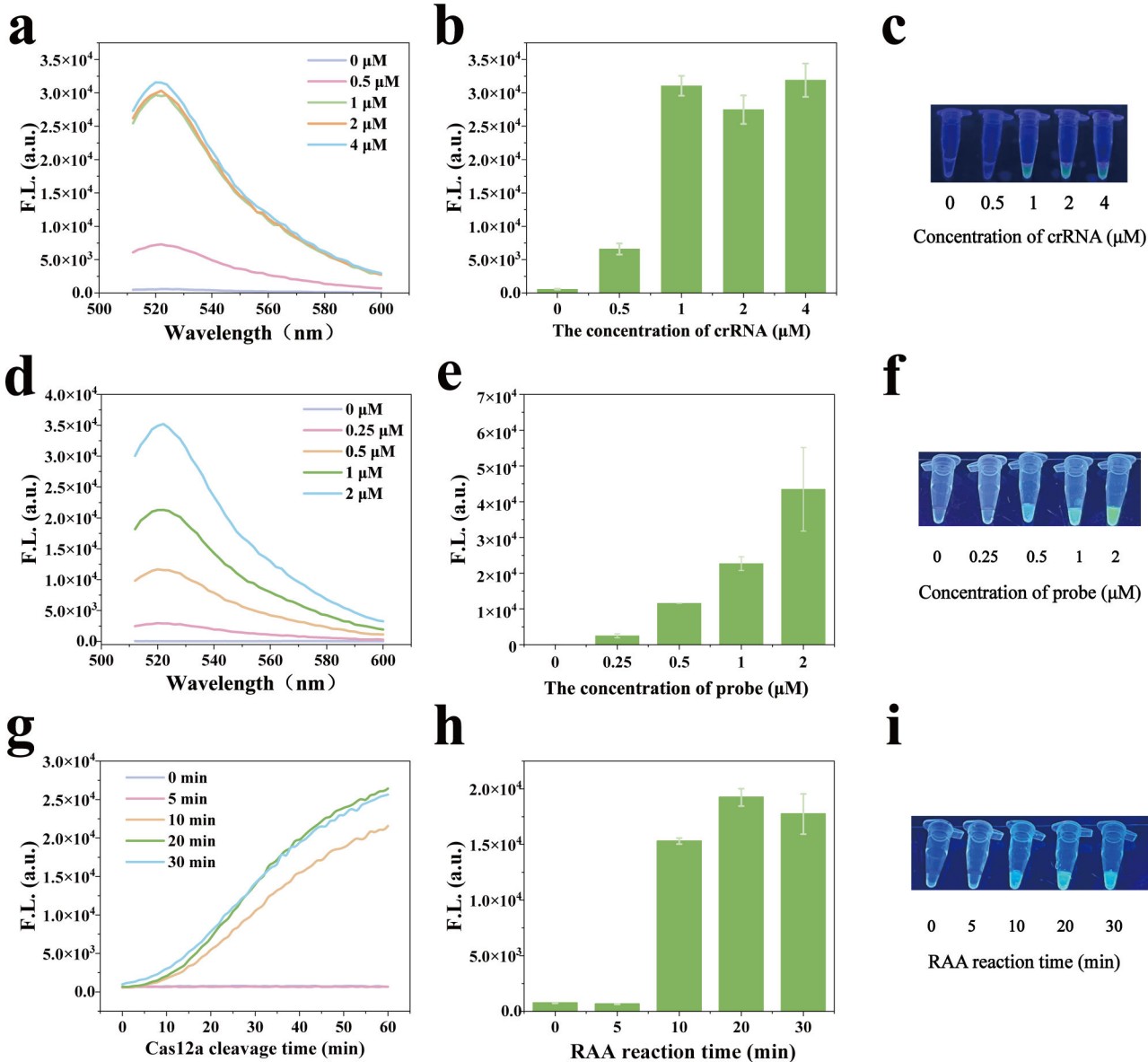

**FIG 3** Optimization of the reaction parameters of the VGRCOT method. (a–c) The concentrations of crRNA on the fluorescence signal. (d–f) The concentrations of ssDNA-Probe on the fluorescence signal. (g–i) The CRISPR/Cas12a cleavage time and RPA time on the fluorescence signal.

c, only GBS can trigger the trans-cutting activity of CRISPR/Cas12a, leading to the fluorescence enhancement at 522 nm, and the same result was obtained by naked-eye observation under UV light. The sensitivity was assessed using varying concentrations of GBS DNA. As shown in Fig. 4d through f, the fluorescence showed a corresponding decrease as the target DNA was reduced from $10^3$ copies/reaction to 0 copies/reaction. Notably, significant fluorescence was observed even at $10^1$ copies/reaction. To confirm the stability of VGRCOT in the low concentration of the target, we selected the $10^1$ copies/reaction group to conduct 10 repetitions (Fig. S4), and the coefficient of variation was 8% (Table S2). The above results indicate that the VGRCOT method has good selectivity, sensitivity, and stability.

## Performance of the VGRCOT method for clinical samples

To further validate the detection performance of the VGRCOT method in clinical application, we assessed 100 samples of genital tract secretions, including 52 negative

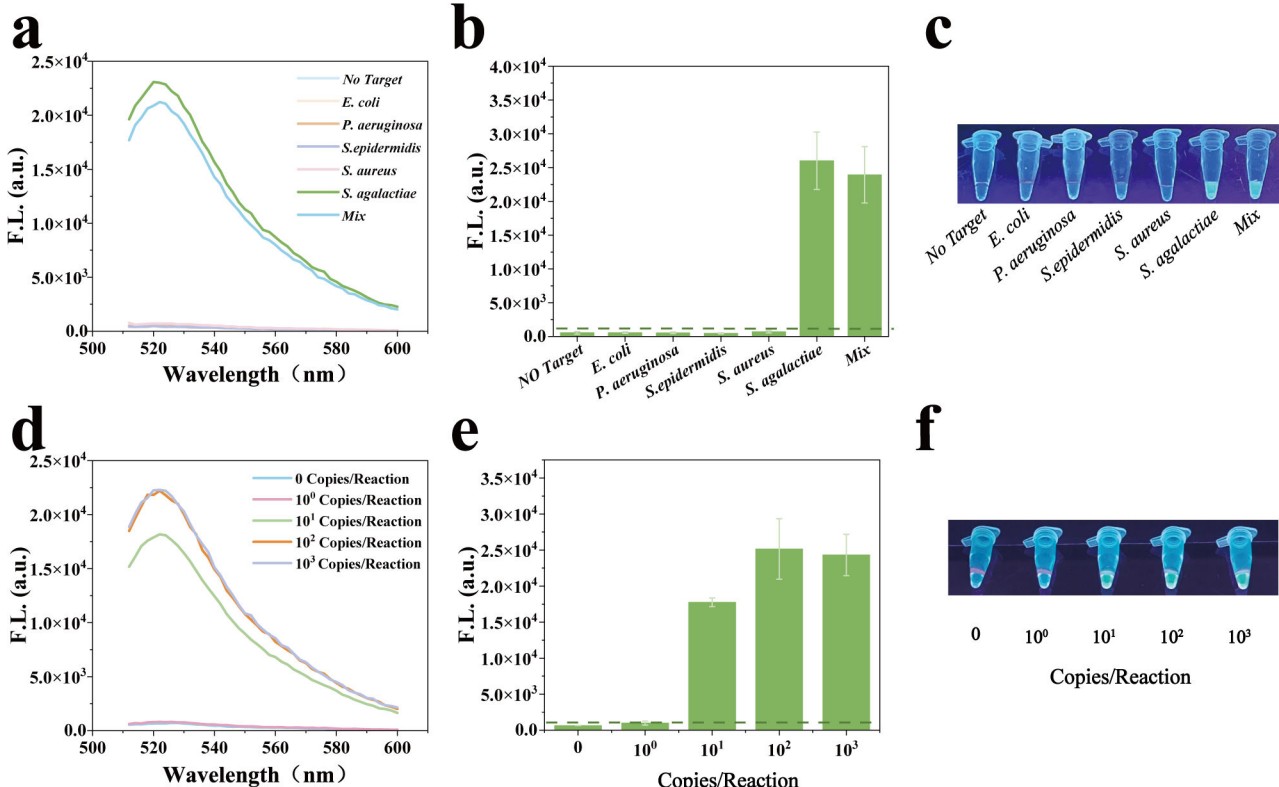

**FIG 4** Specificity and sensitivity of VGRCOT assay in GBS detection. (a) Fluorescence spectra showing the specificity of the VGRCOT method. (b) Results of the statistical histogram of 522 nm fluorescence intensity of specificity analysis by VGRCOT method in GBS detection. (c) Visualized results of specificity analysis in GBS detection. (d) Fluorescence spectra showing the sensitivity of the VGRCOT. (e) Results of the statistical histogram of 522 nm fluorescence intensity of sensitivity analysis by VGRCOT in GBS detection. (f) Visualized results of the sensitivity analysis in GBS detection.

and 48 positive samples. The detection workflow is illustrated in Fig. 5a. Initially, nucleic acids in the genital tract secretion samples were extracted using the thermal lysis method, followed by visual detection using VGRCOT. As shown in Fig. 5b, fluorescence values for all positive samples exceeded the threshold line, while those for negative samples fell below it. Furthermore, we confirmed the detection outcomes of clinical samples by qPCR. The plot shown in Fig. 5c presents the distribution of Ct values for Negative and Positive samples. The Ct values of positive samples ranged from 18 to 34, indicating varying concentrations of the target. The results indicated that VGRCOT and qPCR tests were consistent for all 52 negative samples. Among the 48 positive samples, one case showed a negative result with VGRCOT but a positive result with qPCR (Fig. 5d). The sensitivity, specificity, positive predictive value (PPV), and negative predictive value (NPV) of the VGRCOT were 97.9%, 100.0%, 100.0%, and 98.1%, respectively. These results further substantiate the robust detection capabilities of the VGRCOT method for the identification of GBS.

## Conclusions

GBS poses a significant threat to newborns and pregnant women. Clinical studies have shown that the positive rate of GBS in pregnant women ranges from 5% to 30%, with approximately half of infected women transmitting the bacteria to their newborns (17, 18). It is the leading cause of perinatal infection and neonatal death, seriously threatening the lives and health of pregnant women and their newborns (19, 20). Consensus guidelines advise screening all pregnant women for GBS between the 35th and 37th weeks of pregnancy to reduce the risk of perinatal infection (21). Effective intrapartum antibiotic prophylaxis is crucial for preventing GBS infection in carriers, significantly reducing the incidence and mortality of GBS infection in pregnant women and newborns

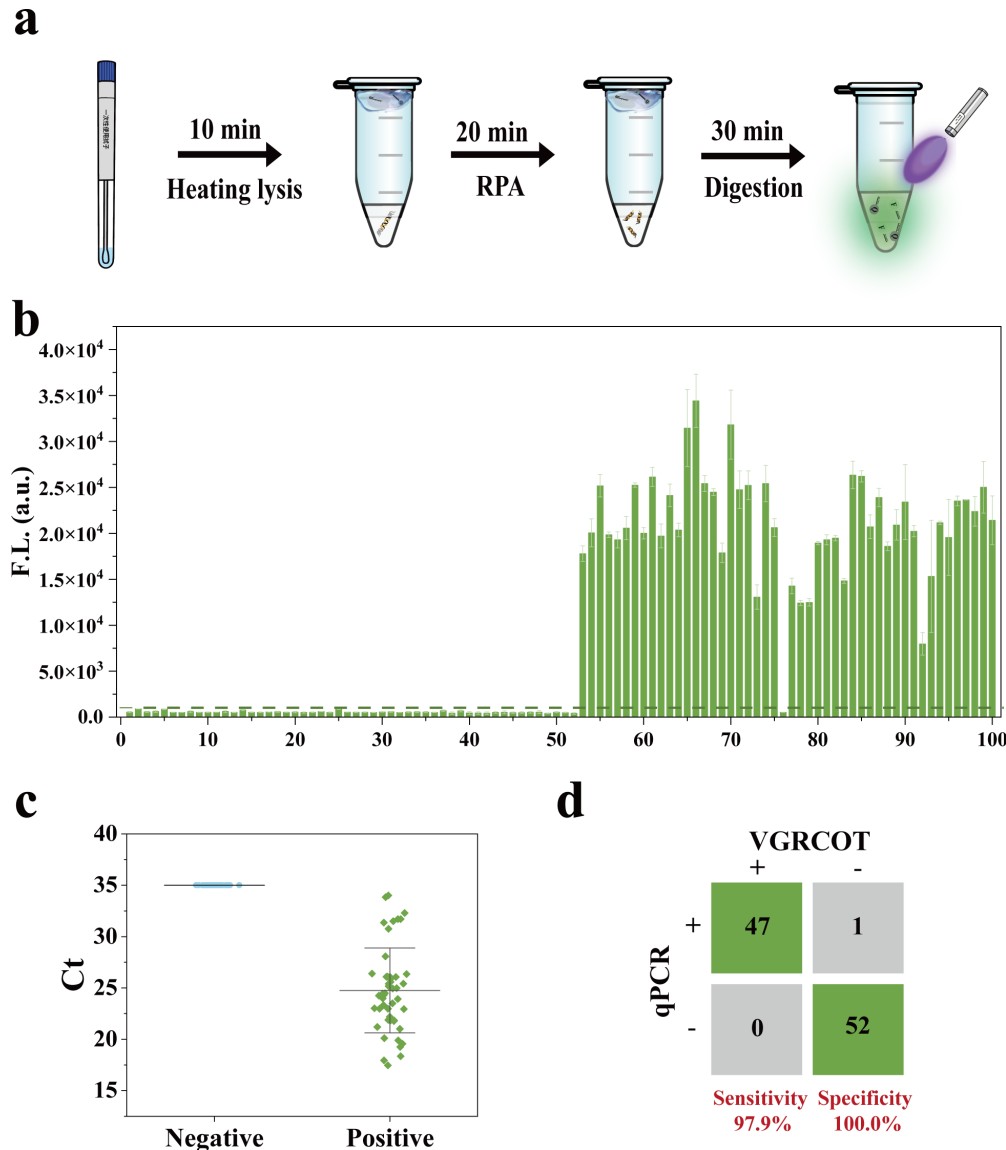

**FIG 5** Performance of the VGRCOT method for clinical samples. (a) Schematic illustration of the workflow of the VGRCOT method for clinical samples. (b) Results of the statistical histogram of 522 nm fluorescence intensity of clinical samples by VGRCOT method. (c) qPCR for the detection of clinical samples. (d) Concordance tables between the VGRCOT and qPCR for 100 clinical specimens, including 52 negative and 48 positive samples.

(22, 23). Therefore, a convenient and visual GBS detection method is essential for controlling GBS infection.

Common methods for detecting GBS in clinical settings include direct culture, immunological tests, and molecular biological methods (24, 25). Direct culture is the gold standard for GBS detection; however, it has a long culture cycle, typically requiring 2–3 days to complete. Factors such as sample quality, specimen collection timing, and culture conditions can affect the success rate of cultures (3). Immunological methods like latex agglutination and colloidal gold immunochromatography have limitations, including long antibody preparation time, low sensitivity, complex procedures, and high false-positive rates (26). Although qPCR technology offers high sensitivity and specificity, it requires sophisticated and expensive equipment, as well as strict requirements for the laboratory environment and operator expertise (27).

The CRISPR/Cas12 system, with its trans-cleavage activity and minimal off-target effects, has significant potential in nucleic acid detection (28). The classical DETECTR

and HOLMES detection systems employ this principle through the use of a fluorescent probe (29, 30). Donghong et al. developed a rapid and sensitive GBS detection method utilizing CRISPR and fluorescent probes (16). However, this method involves stepwise detection for pre-amplification and trans-cleavage processes, which may cause nucleic acid contamination and extend detection times (31). Moreover, its signal collection relies on complex instrumentation, hindering its suitability for on-site detection applications.

In this paper, we developed a one-tube fluorescence visualization assay, VGRCOT, by isolating the RPA amplification and CRISPR/Cas12a system within the bottom and cap of EP tubes, respectively. VGRCOT is straightforward and convenient, requiring only a simple centrifugation step to mix two systems and complete the test, and requires only simple instruments (Fig. S5). By meticulously optimizing the ssDNA-FQ reporter concentration and crRNA concentration, along with the RPA reaction and CRISPR/Cas12a cleavage times, the VGRCOT method achieves rapid visual detection. The entire inspection process takes only 60 minutes, with simple signal output allowing for direct visual interpretation of results without complex instrumentation. VGRCOT exhibits high specificity, clearly distinguishing GBS from other control strains. Finally, testing with 100 clinical specimens confirmed VGRCOT's effective detection performance, which further demonstrates VGRCOT's utility in reproductive health applications.

VGRCOT offers several key advantages that enhance its practicality and diagnostic potential. First, the one-tube visualization method effectively prevents aerosol contamination. The RPA technique is a highly efficient isothermal amplification method that can amplify single-copy nucleic acid templates to detectable levels in a short time. Due to its high amplification efficiency, it is highly susceptible to laboratory contamination in practical use. The CRISPR/Cas12a and RPA components are positioned at the cap and bottom of the EP tube, which eliminates the need to open the lid, thereby reducing aerosol contamination. In addition, after RPA amplification, the two components are mixed through a brief centrifugation step. This setup is simple to operate and physically separates the two reactions, preventing sensitivity loss caused by target competition between CRISPR/Cas12a cleavage and RPA amplification. It achieves a high sensitivity with a detection limit of $10^1$ copies per reaction. Moreover, VGRCOT enables visualized detection without requiring complex instruments, which is more suitable for wards, clinics, or resource-limited areas. Finally, the editable nature of crRNA allows VGRCOT to be easily reprogrammed for detecting various pathogens, demonstrating its strong scalability and broad applicability. In addition, the use of commonly available laboratory reagents and instruments makes VGRCOT support batch production, facilitating low-cost manufacturing.

However, VGRCOT has some limitations, as placing the RPA amplification system and the CRISPR/Cas12a system only on the cap and bottom of the tube does not ensure their separation. During actual operation, meticulous handling is required. Thus, future efforts will explore integrating VGRCOT with microfluidics to enhance the system's sophistication (32, 33). Additionally, the inherent background fluorescence of the probe may impact the accuracy of results in low-concentration samples (34). Addressing the influence of this background fluorescence to more precisely identify low-concentration positives remains a critical challenge.

In conclusion, our VGRCOT method simultaneously satisfies convenience, specificity, sensitivity, and visual detection of GBS. The one-tube, visualization-based design of VGRCOT avoids aerosol contamination and facilitates the interpretation of the test results, offering a novel approach for the rapid detection of GBS. Additionally, due to the programmability of CRISPR/Cas12a, adjusting the crRNA sequence allows VGRCOT to be easily adapted for detecting other pathogenic bacteria. This feature makes this method promising for clinical, on-site, rapid, and early detection of pathogenic microorganisms.

## ACKNOWLEDGMENTS

This work was supported by the Sichuan Provincial Science and Technology Department (2025ZNSFSC1561) and the Key Research and Development Guidance Projects of Deyang City (2024SZY001).

The authors declare that they have no conflict of interest in the publication and no competing interests exist.

All authors consent to publication.

## AUTHOR AFFILIATIONS

[1]Department of Clinical Laboratory, People's Hospital of Deyang City, Chengdu University of Traditional Chinese Medicine, Deyang, Sichuan, China
[2]Department of Clinical Laboratory, Jincheng People's Hospital, Jincheng, Shanxi, China
[3]Department of Clinical Laboratory, Jincheng Hospital Affiliated to Changzhi Medical College, Jincheng, Shanxi, China
[4]Key Laboratory of Laboratory Medicine, School of Laboratory Medicine and Life Sciences, Ministry of Education Wenzhou Medical University, Wenzhou, Zhejiang, China
[5]Digestive Diseases Center, People's Hospital of Deyang City, Chengdu University of Traditional Chinese Medicine, Deyang, Sichuan, China

## AUTHOR ORCIDs

Gang Mai http://orcid.org/0000-0002-5843-2821
Laibao Zheng http://orcid.org/0000-0003-0444-8163
Yayun Jiang http://orcid.org/0009-0008-8632-866X

## FUNDING

| Funder | Grant(s) | Author(s) |
| --- | --- | --- |
| Natural Science Foundation of Sichuan Province | 2025ZNSFSC1561 | Yayun Jiang |
| Key Research and Developmet Guidance Projects of Deyang City | 2024SZY001 | Yayun Jiang |

## AUTHOR CONTRIBUTIONS

Caixia Ji, Methodology | Liqiang Ru, Methodology, Software | Tiao Han, Methodology | Gang Mai, Resources | Laibao Zheng, Writing – review and editing | Yayun Jiang, Funding acquisition, Writing – original draft

## DATA AVAILABILITY

The data sets used and/or analyzed during the current study are available from the corresponding author on reasonable request. The *cfb* gene during the current study is available in the NCBI repository (GenBank accession No GU217532).

## ETHICS APPROVAL

The study was officially approved by the Medical Ethics Committee of the People's Hospital of Deyang City. The ethical review number is 2022-04-088-K01.

## ADDITIONAL FILES

The following material is available online.

## Supplemental Material

**Supplemental material (Spectrum01395-25-S0001.docx).** Fig. S1 to S5; Tables S1 and S2.

## Open Peer Review

**PEER REVIEW HISTORY (review-history.pdf).** An accounting of the reviewer comments and feedback.

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
