## [Reviewer comments · Microbiology Spectrum]

Microbiology Spectrum

VGRCOT: a one-tube visual detection method for group B *Streptococcus* combining RPA and CRISPR/Cas12a for point-of-care testing in reproductive health

Caixia Ji, Liqiang Ru, Tiao Han, Gang Mai, Laibao Zheng, and Yayun Jiang

Corresponding Author(s): Yayun Jiang, Deyang People's Hospital

Review Timeline:

Submission Date:	May 5, 2025
Editorial Decision:	May 29, 2025
Revision Received:	June 13, 2025
Accepted:	June 18, 2025

Editor: Benjamin Liu

Reviewer(s): Disclosure of reviewer identity is with reference to reviewer comments included in decision letter(s). The following individuals involved in review of your submission have agreed to reveal their identity: Reza Azizian (Reviewer #2); tahereh navidifar (Reviewer #3)

Transaction Report:

DOI: <https://doi.org/10.1128/spectrum.01395-25>

Re: Spectrum01395-25 (**VGRCOT: a one-tube visual detection method for group B *Streptococcus* combining RPA and CRISPR/Cas12a for point-of-care testing in reproductive health**)

Dear Dr. Yayun Jiang:

Thank you for the privilege of reviewing your work. Below you will find my comments, instructions from the Spectrum editorial office, and the reviewer comments.

Editor's comments:

1. Lines 51-53: "However, none fulfills the clinical need for convenient, precise, and convenient GBS infection detection due to the long detection period, low sensitivity, and complex instrumentation.": There are no references to support this statement. What's worse, this statement is outdated. Per this new reference (Liu, B.M. Isothermal nucleic acid amplification technologies and CRISPR-Cas based nucleic acid detection strategies for infectious disease diagnostics. In Manual of Molecular Microbiology; ASM Press: Washington, DC, USA, 2026. <https://doi.org/10.1002/9781683674597.ch3>), there are several FDA-approved molecular testing for GBS. One example is Alethia group B *Streptococcus* DNA Amplification Assay (formerly, the Illumigene brand) using loop-mediated isothermal amplification. Besides, the Solana GBS assay is an in vitro diagnostic test that utilizes helicase-dependent amplification for the direct, qualitative detection of GBS from vaginal/rectal swabs following 18 to 24 hours incubation in LIM or Carrot broth cultures. The authors should add these points in the introduction. More references should be cited, with this one (Liu, B.M. Isothermal nucleic acid amplification technologies and CRISPR-Cas based nucleic acid detection strategies for infectious disease diagnostics. In Manual of Molecular Microbiology; ASM Press: Washington, DC, USA, 2026. <https://doi.org/10.1002/9781683674597.ch3>) as an example (citing is optional).

2. In Introduction, the authors should justify why they chose CRISPR/Cas system for the detection of GBS. They should compare the pros and cons of CRISPR/Cas system and other POCT methods. More references should be cited, with this one (PMID: 39857007) as an example (citing is optional).

Please return the manuscript within 30 days; if you cannot complete the modification within this time period, please contact me. If you do not wish to modify the manuscript and prefer to submit it to another journal, notify me immediately so that the manuscript may be formally withdrawn from consideration by Spectrum.

Revision Guidelines

ASM Membership: Corresponding authors may join or renew ASM membership to obtain discounts on publication fees. Need

to upgrade your membership level? Please contact Customer Service at Service@asmusa.org.

Sincerely,
Benjamin Liu
Editor
Microbiology Spectrum

Reviewer #2 (Comments for the Author):

There is a need to clarify the lot or catalog number for all reagents and kits used in our research to ensure reproducibility and transparency of results. Additionally, if available, please provide the ATCC number for the standard strains of *Streptococcus* used in this study.

Furthermore, to provide a comprehensive comparison with qPCR, it would be beneficial to include details on the performance characteristics of the RPA methods, such as the positive predictive value (PPV), negative predictive value (NPV), limit of detection (LOD), and limit of quantitation (LOQ). Such information will enable a more accurate assessment of the RPA methods. And also, there is a need for clarification of ethical codes.

Reviewer #3 (Comments for the Author):

"VGROOT elegantly combines RPA and CRISPR/Cas12a into a onetube assay, offering a leap forward in pointofcare GBS detection. Addressing background fluorescence and scalability will solidify its translational impact."

1. Strengths of the Study:

Innovative Methodology:

OneTube Design: Novel integration of RPA and CRISPR/Cas12a in a single tube minimizes contamination risk and simplifies workflow, addressing a critical limitation of existing twostep methods.

Visual Detection: Fluorescence readout under UV light enables instrumentfree interpretation, ideal for lowresource settings.

Technical Rigor:

Optimization: Systematic optimization of key parameters (crRNA concentration, ssDNAFQ reporter, reaction times) ensures robust performance.

Sensitivity/Specificity: Achieves a low detection limit (10^1 copies/reaction) and distinguishes GBS from nontarget bacteria (e.g., *E. coli*, *S. aureus*).

Clinical Validation:

qPCR Concordance: Results from 30 clinical samples align with qPCR, demonstrating reliability.

Practicality: Short turnaround time (60 min) and minimal equipment (centrifuge, UV light) enhance pointofcare applicability.

Contamination Control:

Physical separation of RPA and CRISPR/Cas12a systems in the tube cap/base reduces aerosol contamination risks.

2. Limitations and Concerns:

△ Mechanical Reliability:

The "onetube" design relies on manual centrifugation to mix reagents, which may introduce variability. A microfluidic integration (as noted in the limitations) could standardize this step.

△ Background Fluorescence:

The ssDNAFQ reporter's inherent background fluorescence (mentioned in Line 366-369) may affect lowconcentration sample accuracy. Quantifying this noise (e.g., signaltonoise ratio) would strengthen claims.

△ Sample Preparation:

Nucleic acid extraction via thermal lysis (Line 154) is less robust than columnbased methods. Potential inhibitors in clinical samples (e.g., vaginal secretions) could impact sensitivity.

△ Scalability:

The method's dependency on precise reagent placement in tube caps/bases may hinder mass production or automation.

△ Comparative Performance:

No direct comparison with commercial GBS tests (e.g., Cepheid Xpert GBS) to benchmark cost, speed, or ease of use.

3. Recommendations for Improvement:

◆ Address Background Fluorescence:

Include a negative control baseline in fluorescence intensity plots (e.g., Figure 4d-f) to clarify detection thresholds.

Test alternative reporters (e.g., quenched probes) to reduce noise.

◆ Enhance Robustness:

Validate with a larger clinical cohort (e.g., >100 samples) across diverse populations.

Assess performance with minimally processed samples (e.g., direct swab lysates) to simulate pointofcare use.

◆ Microfluidic Integration:

Prototype a sealed, disposable cartridge (as suggested in Lines 364-366) to automate mixing and improve reproducibility.

◆ Comparative Analysis:

Compare VGROOT's cost, time, and accuracy against existing POCTs (e.g., Alere i GBS) to highlight advantages.

4. Overall Assessment:

Impact: High potential for GBS screening in resource-limited settings due to rapid, visual, and contamination-resistant design.

Innovation: 9/10

Technical Soundness: 8/10

Clinical Relevance: 8/10

Decision: Minor Revisions (address limitations above) before final acceptance.

Suggested Revisions:

1. Quantify background fluorescence and propose mitigation strategies.
2. Expand clinical validation with larger/diverse cohorts.
3. Discuss scalability and manufacturing feasibility.

Comments and Suggestions for the Author:

There is a need to clarify the lot or catalog number for all reagents and kits used in our research to ensure reproducibility and transparency of results. Additionally, if available, please provide the ATCC number for the standard strains of *Streptococcus* used in this study.

Furthermore, to provide a comprehensive comparison with qPCR, it would be beneficial to include details on the performance characteristics of the RPA methods, such as the positive predictive value (PPV), negative predictive value (NPV), limit of detection (LOD), and limit of quantitation (LOQ). Such information will enable a more accurate assessment of the RPA methods. And also, there is a need for clarification of ethical codes.

Spectrum01395-25 (**VGRCOT: a one-tube visual detection method for group B *Streptococcus* combining RPA and CRISPR/Cas12a for point-of-care testing in reproductive health**)

Dear Editor:

We gratefully thank you for spending time to remark useful suggestions, which have significantly raised the quality of the manuscript and enabled us to improve the manuscript. Each suggested revision and comment was accurately incorporated and considered. Reviewer comments are shown in black and our response in blue. In the revised manuscript changes are marked in red.

Editor's comments:

1. Lines 51-53: "However, none fulfills the clinical need for convenient, precise, and convenient GBS infection detection due to the long detection period, low sensitivity, and complex instrumentation." There are no references to support this statement. What's worse, this statement is outdated. Per this new reference (Liu, B.M. Isothermal nucleic acid amplification technologies and CRISPR-Cas based nucleic acid detection strategies for infectious disease diagnostics. In *Manual of Molecular Microbiology*; ASM Press: Washington, DC, USA, 2026.<https://doi.org/10.1002/9781683674597.ch3>), there are several FDA-approved molecular testing for GBS. One example is Alethia group B *Streptococcus* DNA Amplification Assay (formerly, the Illumigene brand) using loop-mediated isothermal amplification. Besides, the Solana GBS assay

is an in vitro diagnostic test that utilizes helicase-dependent amplification for the direct, qualitative detection of GBS from vaginal/rectal swabs following 18 to 24 hours incubation in LIM or Carrot broth cultures. The authors should add these points in the introduction. More references should be cited, with this one (Liu, B.M. Isothermal nucleic acid amplification technologies and CRISPR-Cas based nucleic acid detection strategies for infectious disease diagnostics. In Manual of Molecular Microbiology; ASM Press: Washington, DC, USA, 2026.<https://doi.org/10.1002/9781683674597.ch3>) as an example (citing is optional).

Response: Thanks for pointing out this important issue. In Lines 51-53, our description is not accurate enough and does not reflect the latest advances in GBS detection technology. In the revised manuscript, we have deleted it and added some introductions to FDA-approved molecular testing for GBS to more accurately reflect the current status of GBS testing. We thoroughly read the paper titled “Isothermal Nucleic Acid Amplification Technology and CRISPR-Cas Based Nucleic Acid Detection Strategy for Infectious Disease Diagnosis.” The article is closely related to our research and has enhanced my understanding of the relevant background. To increase its visibility, we cited it as a reference. The updated version is as follows:

Currently, bacterial culture, immunoassay, and Polymerase chain reaction (PCR) are commonly used for the clinical detection of GBS(4, 5). Although bacterial culture is considered the gold standard for diagnosing GBS, it is cumbersome and time-consuming. Immunoassays, which rely on specific antibody–antigen interactions, are rapid but have low sensitivity. PCR offers high sensitivity and specificity;

however, it requires sophisticated equipment, skilled personnel, and a laboratory environment. Several FDA-approved molecular diagnostic tests for GBS have been developed in recent years(6). One example is the Alethia GBS DNA Amplification Assay, which utilizes loop-mediated isothermal amplification (LAMP) technology. Another is the Solana GBS assay, an in vitro diagnostic test that employs helicase-dependent amplification (HDA) for the direct, qualitative detection of GBS from vaginal or rectal swabs after 18 to 24 hours of incubation in carrot broth cultures. However, these methods also have certain limitations in point-of-care (POC) testing.

2. In introduction, the authors should justify why they chose CRISPR/Cas system for the detection of GBS. They should compare the pros and cons of CRISPR/Cas system and other POCT methods. More references should be cited, with this one (PMID: 39857007) as an example (citing is optional).

Response: Thanks for pointing out this important issue. We do lack relevant introduction in this section. Here, we have added a comparison between the pros and cons of CRISPR/Cas system and other POCT methods in this section, citing the paper "Impact of Point-of-Care Testing on Diagnosis, Treatment, and Surveillance of Vaccine-Preventable Viral Infections" to enhance the clarity of the article. The updated version is as follows:

POC testing is critical in monitoring infectious diseases, tracking disease progression, and responding to outbreaks. It is especially valuable in resource-limited settings and offers significant advantages for public health strategies(7). Various POC

testing methods have been developed, with the most notable including lateral flow assays (LFA), biosensor-based assays, and Clustered regularly interspaced short palindromic repeats (CRISPR)-based assays(8). LFA provides a rapid, cost-effective, and portable approach suitable for initial screening. However, its relatively low sensitivity and specificity reduce diagnostic accuracy, especially when sampling quality is suboptimal(9). Biosensor-based assays offer high sensitivity and specificity with a portable design, but biorecognition components can be costly, and the results may be influenced by environmental factors(10).

CRISPR/CRISPR-associated protein (Cas) systems are acquired immune systems formed by bacteria and archaea during biological evolution to defend against invasion by exogenous substances such as phages and plasmids. The CRISPR/Cas system has the function of accurately identifying and cleaving target DNA or RNA sequences, serving as a crucial tool for transcriptional regulation, gene editing, and DNA/RNA detection, acclaimed as the next-generation molecular detection technology(11, 12). Notably, CRISPR/Cas12a can not only specifically identify and cleave target double-stranded DNA (dsDNA) but also indiscriminately cleave single-stranded DNA (ssDNA), demonstrating significant application potential in the POC testing(13). However, several limitations may affect its practicality, including low sensitivity, technical complexity, and risks of false positives due to contamination, all of which hinder its practical application in rapid and low-resource settings(8). Recombinase polymerase amplification (RPA) is an amplification technique that mimics nucleic acid replication processes in vitro, utilizing recombinase,

single-stranded DNA-binding proteins, and strand-displacement DNA polymerase(14). RPA offers high sensitivity, excellent specificity, and easy operation as key advantages, which are widely used in biosensing and are an ideal choice for developing CRISPR/Cas-based methods(15). Some fluorescent assays based on CRISPR/Cas12a and RPA have been constructed, but these mostly require sophisticated instrumentation to collect fluorescent signals and are realized by two separate reactions. Furthermore, the wo-tube method involves uncapping and liquid transfer, exposing the RPA amplification product to air, which increases the risk of laboratory contamination and potentially leads to false positives(16).

Reviewer #2 (Comments for the Author):

1. There is a need to clarify the lot or catalog number for all reagents and kits used in our research to ensure reproducibility and transparency of results. Additionally, if available, please provide the ATCC number for the standard strains of *Streptococcus* used in this study.

Response: Thanks for pointing out this important issue. We have carefully revised the manuscript to include the catalog numbers for all reagents and kits used in this study to enhance the reproducibility and transparency of our results. In addition, the ATCC numbers of all standard strains have been provided, as requested. These details have been incorporated into the revised Methods section. The updated version is as follows:

LbCas12a (Catalog # M0653T) and RNA transcription Kits (Catalog # E2050S) were purchased from New England Biolabs Inc. (United States). RAA FAST (Catalog # B00000), the recombinase polymerase amplification (RPA) nucleic acid amplification kits were obtained from Qitian Gene Biological Co., Ltd. (Jiangsu, China). RNA purification kits (Catalog # R0071S) were purchased from Biyuntian Biotechnology Co. (Shanghai, China). Ultra GelRed (Catalog # GR501-01) and a 100 bp DNA Ladder (Catalog # MD104-01) were provided by Vazyme Co., Ltd. (Nanjing, China). SGExcel Fast SYBR qPCR Mixture (Catalog # B630011), crRNA, and other nucleic acid sequences were synthesized by Sangon Biotech (Shanghai, China). *Escherichia coli* (*E. coli*, ATCC 25922), *Pseudomonas aeruginosa* (*P. aeruginosa*, ATCC 27853), *Staphylococcus epidermidis* (*S. epidermidis*, ATCC 12228),

Staphylococcus aureus (*S. aureus*, ATCC 29213), and *Streptococcus agalactiae* (*S. agalactiae*, ATCC 12386) were laboratory-preserved strains.

2. Furthermore, to provide a comprehensive comparison with qPCR, it would be beneficial to include details on the performance characteristics of the RPA methods, such as the positive predictive value (PPV), negative predictive value (NPV), limit of detection (LOD), and limit of quantitation (LOQ). Such information will enable a more accurate assessment of the RPA methods.

Response: To assess the detection performance of VGRCOT, we increased the number of samples to 100, including 52 negative and 48 positive samples, and evaluated its clinical applicability (Figure 5). The results indicated that VGRCOT and qPCR tests were consistent for all 52 negative samples. Among the 48 positive samples, one case showed a negative result with VGRCOT but a positive result with qPCR. The sensitivity, specificity, positive predictive value (PPV), and negative predictive value (NPV) of the VGRCOT were 97.9%, 100%, 100.0%, and 98.1%, respectively. In Figures 4d-f, we assessed the detection limit of VGRCOT, which showed that it can detect as few as 10^1 copies of target DNA.

Figure 5. Performance of the VGR-COT method for clinical samples (a) Schematic illustration of the workflow of the VGR-COT method for clinical samples. (b) Results of the statistical histogram of 522 nm fluorescence intensity of clinical samples by VGR-COT method. (c) qPCR for the detection of clinical samples. (d) Concordance tables between the VGR-COT and qPCR for 100 clinical specimens, including 52 negative and 48 positive samples.

Finally, I would like to explain that VGR-COT is a qualitative detection method

that aims to determine whether there is GBS infection by visual observation. It is not intended for quantitative detection. Therefore, we did not assess its limit of quantitation (LOQ). The relevant information has been incorporated into the revised manuscript.

3. There is a need for clarification of ethical codes.

Ethics statement: The study was officially approved by the Medical Ethics Committee of the People's Hospital of Deyang City. The ethical review number is 2022-04-088-K01.

Reviewer #3 (Comments for the Author):

"VGROOT elegantly combines RPA and CRISPR/Cas12a into a onetube assay, offering a leap forward in pointofcare GBS detection. Addressing background fluorescence and scalability will solidify its translational impact."

Overall Assessment:

Impact: High potential for GBS screening in resourcelimited settings due to rapid, visual, and contaminationresistant design.

Innovation: 9/10

Technical Soundness: 8/10

Clinical Relevance: 8/10

Decision: Minor Revisions (address limitations above) before final acceptance.

Suggested Revisions:

1. Quantify background fluorescence and propose mitigation strategies.

Response: Thank you for your valuable comment. The background fluorescence of the probe may impact the accuracy of results and lead to false positives. To address this, we calculated the threshold of VGRCOT to quantify the background and determine whether the result is negative or positive. The threshold was performed by 10 repeated experiments with negative samples (Figure S3). The threshold was calculated using the formula $TH = X + 3SD$, where X is the mean signal intensity and SD is the corresponding standard deviation^[1-3]. Specifically, $TH = 783.6 + 3 \times 80.73$

= 1025.79. Therefore, the threshold value of 1025.79 is used to distinguish between positive and negative results. The test's sensitivity is also defined by this threshold. The detailed calculation process has been included in the article.

Figure S3. Calculate the cut-off value for fluorescent outputs using the VGRCOT method. $X + 3SD$ calculates the cut-off value, X represents an average signal value of three replicates without using the target gene, and SD represents three standard deviations.

- [1] G. Yang, S. Yu, Y. Liu, J. Huang, Q. Li, Z.P. Aguilar, H. Xu, A fluorescence-positioned hybridization chain reaction system for sensitive detection of: *Salmonella* in milk, *Anal. Methods*. 12 (2020) 1958–1965. <https://doi.org/10.1039/d0ay00184h>.
- [2] J. Hu, F. Tang, L. Wang, M. Tang, Y.Z. Jiang, C. Liu, Nanozyme sensor based-on platinum-decorated polymer nanosphere for rapid and sensitive detection of *Salmonella typhimurium* with the naked eye, *Sensors Actuators B Chem*. 346 (2021) 130560. <https://doi.org/10.1016/j.snb.2021.130560>.
- [3] F. Li, Q. Ye, M. Chen, B. Zhou, J. Zhang, R. Pang, L. Xue, J. Wang, H. Zeng,

S. Wu, Y. Zhang, Y. Ding, Q. Wu, An ultrasensitive CRISPR/Cas12a based electrochemical biosensor for *Listeria monocytogenes* detection, *Biosens. Bioelectron.* 179 (2021) 113073. <https://doi.org/10.1016/j.bios.2021.113073>.

2. Expand clinical validation with larger/diverse cohorts.

Response: Thank you for your valuable suggestion. We expanded the clinical validation to 100 specimens, consisting of 52 negative and 48 positive samples, thereby improving the robustness and generalizability of our findings. The results indicated that VGRCOT and qPCR tests were consistent for all 52 negative samples. Among the 48 positive samples, one case showed a negative result with VGRCOT but a positive result with qPCR (Figure 5d). The sensitivity, specificity, positive predictive value (PPV), and negative predictive value (NPV) of the VGRCOT were 97.9%, 100.0%, 100.0%, and 98.1%, respectively. These results further substantiate the robust detection capabilities of the VGRCOT method for the identification of GBS.

Figure 5. Performance of the VGR-COT method for clinical samples (a) Schematic illustration of the workflow of the VGR-COT method for clinical samples. (b) Results of the statistical histogram of 522 nm fluorescence intensity of clinical samples by VGR-COT method. (c) qPCR for the detection of clinical samples. (d) Concordance tables between the VGR-COT and qPCR for 100 clinical specimens, including 52 negative and 48 positive samples.

3. Discuss scalability and manufacturing feasibility.

Response: Thank you for your valuable comment. Scalability and manufacturing feasibility are indeed important advantages of VGRCOT. We have revised the discussion section of the manuscript to emphasize these two points. The updated version is as follows:

VGRCOT offers several key advantages that enhance its practicality and diagnostic potential. First, the one-tube visualization method effectively prevents aerosol contamination. The RPA technique is a highly efficient isothermal amplification method that can amplify single-copy nucleic acid templates to detectable levels in a short time. Due to its high amplification efficiency, it is highly susceptible to laboratory contamination in practical use. The CRISPR/Cas12a and RPA components are positioned at the cap and bottom of the EP tube, which eliminates the need to open the lid, thereby reducing aerosol contamination. In addition, after RPA amplification, the two components are mixed through a brief centrifugation step. This setup is simple to operate and physically separates the two reactions, preventing sensitivity loss caused by target competition between CRISPR/Cas12a cleavage and RPA amplification. It achieves a high sensitivity with a detection limit of 10^1 copies per reaction. Moreover, VGRCOT enables visualized detection without requiring complex instruments, which is more suitable for wards, clinics, or resource-limited areas. Finally, the editable nature of crRNA allows VGRCOT to be easily reprogrammed for detecting various pathogens, demonstrating its strong scalability and broad applicability. In addition, the use of commonly

available laboratory reagents and instruments makes VGCOT support batch production, facilitating low-cost manufacturing.

Re: Spectrum01395-25R1 (**VGRCOT: a one-tube visual detection method for group B *Streptococcus* combining RPA and CRISPR/Cas12a for point-of-care testing in reproductive health**)

Dear Dr. Yayun Jiang:

Your manuscript has been accepted, and I am forwarding it to the ASM production staff for publication. Your paper will first be checked to make sure all elements meet the technical requirements. ASM staff will contact you if anything needs to be revised before copyediting and production can begin. Otherwise, you will be notified when your proofs are ready to be viewed.

Sincerely,
Benjamin Liu
Editor
Microbiology Spectrum